# OpenReview forum: "RBCBF: Decoding Time Safety Alignment via Risk Guided Rollback and Barrier Control"
_ICML.cc/2026/Conference — ICML 2026 regular_

### Official Review · Reviewer_uWVe · 2026-03-07

**Soundness:** 3
**Presentation:** 3
**Significance:** 3
**Originality:** 3
**Overall Recommendation:** 4
**Confidence:** 4

**Summary:**

I see that authors introduce RBCBF, a decoding-time safety method that works in two parts. First, if the safety evaluator detects a potential rule violation, the system rolls back to an earlier prefix in the generation. Second, it adjusts the token distribution using a control-inspired constraint: essentially a KL-based projection onto a restricted simplex, with added forcing and slack terms so the projection still works under top-k or top-p truncation. The main novelty they highlight is how they choose rollback points-they use a 'risk concentration' signal based on risk accumulation or negative drift, rather than relying on simple heuristics.

**Compliance With Llm Reviewing Policy:**

Affirmed.

**Final Justification:**

Authors have address some of the concerns

**Key Questions For Authors:**

What do the end-to-end results look like across the full prompt set, not just the cases where the safety scorer triggered? It would help to see the unconditional safety and helpfulness numbers.

How robust is RBCBF to scorer errors or mismatches between the generator and the safety scorer? Some analysis of scorer dependence would clarify how stable the method really is.

Can you compare RBCBF against stronger and better-tuned decoding-time safety baselines under matched compute budgets? That would make the baseline comparisons feel more fair.

What is the actual wall-clock overhead of running KL-CBF control, and how does it scale with model size, decoding length, or evaluator frequency?

And finally, does the 'risk concentration' signal genuinely identify meaningful rollback points, or does it mostly behave like another heuristic windowing rule? Some validation here would be useful.

**Limitations:**

No, They do acknowledge some limitations (dependence on the scorer/rules; potential instability in localization; vulnerability to escape attacks), but the discussion is not yet adequate given the safety/security positioning.

**Strengths And Weaknesses:**

Strengths:

I think the conceptual framing for choosing rollback points is solid. The idea that 'risk accumulates along the prefix' feels realistic, and using the negative-drift signal along with the α-coverage/compact-window setup is a sensible way to pinpoint where things started to go wrong, instead of relying on arbitrary buffer sizes or just rolling back to the last triggered step.

The formulation for adjusting the token distribution is also clean. Treating the correction as a KL-projection under linear constraints makes the method easy to understand and analyze. The discussion around feasibility-especially the convex-hull dominance and slack relaxation-adds clarity that many decoding-time approaches gloss over.

I appreciate that the method explicitly deals with truncated decoding. Since most real LLM decoding isn't full-vocab, acknowledging the feasibility issues that come from top-k/top-p truncation-and building slack into the method rather than tacking it on-is a thoughtful design choice.

The ablations are helpful as well. The sensitivity tests on window length and the force/slack variants show real behavioral differences, which suggests the method is doing more than just adding cosmetic constraints.

Weakness:

I feel the 'safety guarantees' come across stronger than what the method can actually deliver. The whole setup depends on a noisy learned evaluator (sometimes not even the same model as the generator), the generation process is stochastic, constraints only apply on the truncated token set, and slack means violations are allowed when needed. So the guarantee is really 'in expectation, according to the evaluator,' which is very different from the hard guarantees CBFs usually imply. It would help if the paper were more upfront that this is a soft, evaluator-relative safety mechanism.

The approach also leans heavily on the safety scorer and the evaluation protocol. Many of the gains could simply be artifacts of the specific rule set, scorer prompts, gating logic, or the choice to only look at cases that triggered the safety scorer. Studying triggered cases is fair, but it can easily make the method look more effective than it actually is on the full prompt distribution. It would be helpful to see complete end‑to‑end metrics-how often triggers happen, refusal rates, and the overall harm rate across all prompts-along with scorer calibration or error analysis.

The baseline comparisons feel a bit underpowered. Rollback-Introspect is implemented using a minimal template, and logit steering is known to depend heavily on tuning. If the paper claims to outperform prior rollback methods, I think it needs stronger baselines, clearer tuning parity, and ideally tests on more than one base model.

The 'risk concentration' procedure, while presented as principled, still depends on a lot of hyperparameters - activation thresholds, α‑coverage ratios, pre-trigger window sizes, peak-detector settings, and the mapping policy. None of these are unreasonable, but taken together it makes the method feel more like a structured heuristic pipeline. It would help to see how stable rollback selection is under small variations in scorer behavior or changes to these knobs.

The quality evaluation isn't very strongly grounded. Using a small LLM judge for win-rates is common, but pretty fragile-especially in safety work where 'quality' can collapse into 'refusal style.' Some human evaluation, task-based usefulness checks, or at least larger judges with inter-judge agreement would make the claims more convincing.

And on the practical side, I think the compute overhead could use more clarity. The real cost depends on how often the evaluator runs, how expensive the token-level risk increments are, the cost of solving the KL-projection step, and how frequently rollbacks actually occur. The unified 'Cost' metric is helpful, but it doesn't fully capture the operational reality.

---

> ### Author Rebuttal · Authors · 2026-03-31
>
> We thank the reviewer for the insightful comments and for the appreciation of the mathematical framework and experiments.
>
> **1. (Q1 & W2 & W5)**
>
> **Table 3. Full-Distribution WJB Evaluation**
> *Generator: Qwen2.5-7B-Instruct.*
>
> | Group       | Type        | Trigger (RBCBF) | Trigger (NoCtl) | $\Delta h_x(t^* \to F)$ | $h(x_F) > \epsilon$@triggered |
> | :---------- | :---------- | --------------: | --------------: | ----------------------: | ----------------------------: |
> | **Harmful** | adv_harmful |           82.1% |           80.4% |                   +5.93 |                         85.2% |
> |             | van_harmful |           81.2% |           83.3% |                   +4.76 |                         86.4% |
> |             | **avg**     |       **81.7%** |       **81.9%** |               **+5.35** |                     **85.8%** |
> | **Benign**  | adv_benign  |           47.5% |           45.3% |                       — |                             — |
> |             | van_benign  |           23.2% |           21.0% |                       — |                             — |
> |             | **avg**     |       **35.4%** |       **33.2%** |                   **—** |                         **—** |
>
> We evaluate RBCBF over the full prompt distribution, comparing it with NoControl mode. The results are shown in Table 3. We use 800 WJB prompts (200 per category, covering adversarial/vanilla × harmful/benign). RBCBF improves performance on harmful prompts while preserving behavior on benign prompts. For the two harmful categories, the average trigger rate is 81.7% for RBCBF and 81.9% for NoControl. This indicates that trigger frequency is mainly dominated by the scorer. Conditioned on triggered harmful cases, RBCBF improves terminal safety. The average $\Delta h_x(t^* \to F)$ is +5.35, and 85.8% of triggered harmful generations return to the safe region. Second, we report the nonzero trigger rate on benign prompts. NoControl exhibits nontrivial benign trigger rates (45.3% and 21.0%). These benign triggers likely reflect the strictness of the scoring template and scorer calibration. Overall, the results are consistent with expectations. As attack intent weakens, the trigger rate decreases across the four prompt subsets.  For additional evidence on scorers and generators, please see **1. (W2 & Q4: scorer dependence)** in our response to **Reviewer rxfS**, as well as **Table 2: Generator Generalizability** under **1. (W1 & W4)** in our response to **Reviewer Pdb7**.
>
> **2. (W3 & Q3)**
>
> We compare RBCBF against two representative decoding-time safety baselines. The only difference is the post-rollback mechanism where RBCBF applies KL-CBF projection, and INTROSPECT regenerates by following the template. For logit steering, we follow the original paper and its open-source setup as closely as possible, without adding extra tuning on our side. We believe this is enough here, because our goal is to compare the safety mechanism itself, rather than gains from additional tuning effort. Furthermore, we intentionally do not use a complex regeneration template. In order to keep the attribution clean. Otherwise, we would introduce a second variable, namely the model’s ability to follow template rules, making it difficult to distinguish whether the gain comes from (1) the rollback structure itself or (2) the prompt/template design. In that case, the comparison would become closer to prompt engineering than to evaluating risk concentration or KL-CBF itself. Since this paper focuses on how CBF theory benefits safety alignment, we believe the current experimental design and the additional rebuttal results are sufficient to support the contribution of RBCBF.
>
> **3. (W6 & Q4)**
>
> We refer the reviewer to **2. (W2 & Q2: Wall-clock)** in our response to **Reviewer ZLtM**, where we provide a more concrete discussion of the practical wall-clock overhead.
>
> **4. (Q2)**
>
>  We refer the reviewer to **Table 7: Scorer Noise Robustness of RBCBF** based on **1. (W1 & W4)** in our response to **Reviewer Pdb7**.
>
> **5. (W4 & Q5)**
>
> We believe that $t^\*$ is not a fixed-window heuristic. Across all triggered sequences, the trigger position spans 0 to 107 tokens (CV = 1.32), indicating substantial temporal dispersion. Furthermore, the risk concentration signal identifies semantically meaningful rollback points. At $\sigma = 0$, we observe $\Delta h(t^* \to F) = +5.87$ (Table 6), and under 4x scorer noise (Table 7), it remains stable from 4.93 to 5.30, demonstrating robustness to noise. In addition, across different generators, the variation of $h(x_{t^*})$ is only 0.30 (Table 2). These results indicate that the trigger point aligns with semantic content. In other words, risk concentration can reliably capture appropriate rollback points.
>
> **6. (W1)**
>
> We agree that the term “safety guarantee” may be too strong, and will revise the manuscript to present RBCBF as an evaluator-relative safety mechanism rather than a hard guarantee.

---

> > ### Author Rebuttal · Reviewer_uWVe · 2026-04-03
> >
> > Thank you for your response and the rebuttal helps most on full-distribution evaluation and gives some support for rollback localization, but it does not fully resolve the concerns about scorer dependence, baseline strength, compute overhead, or the heuristic nature of the rollback-selection pipeline. I'm providing below my verdict and please let me know, If you like me to provide more insights
> >
> > Q1: partially responded
> >
> > Q2: partially responded
> >
> > Q3: not adequately responded
> >
> > Q4: not adequately responded
> >
> > Q5: partially responded
> >
> > I choose to remain on my proposed score.

---

> > > ### Author Response · Authors · 2026-04-04
> > >
> > > We thank the reviewer for the follow-up. We would greatly appreciate it if the reviewer could **clarify where the remaining confusion lies**, so that we can provide a more targeted response. Note that due to the 5,000-character limit, our previous reply used cross-references for overlapping questions, per the ICML 2026 guidelines.
> > >
> > > **1 (Q1)**
> > > We have **provided end-to-end results in our previous rebuttal across the full prompt set in Table 3**, which, we believe, should address the reviewer’s concerns. We would be happy to clarify further if there is any specific unsolved point.
> > >
> > > **2 (Q2)**
> > > Let us restate the key points about how robust RBCBF is to scorer errors or mismatches between the generator and the safety scorer. First, we explain the **scorer dependence** in our response to **Reviewer rxfS** under **W2 & Q4: scorer dependence** (see Table 1). Second, we clarify the **Scorer Noise Robustness** in our response to **Reviewer Pdb7 in Table 7**. Third, we report the **Generator Generalizability** based on **Table 2 in our response to Reviewer Pdb7**. We would be happy to clarify further if there is any specific unsolved point.
> > >
> > > **3 (Q3)**
> > > We first clarify that, **beyond the two main baselines (LOGIT-STEER and ROLLBACK-INTROSPECT), we have also compared RBCBF against two additional hard-filtering CBF baselines, HardFilter-AlwaysOn and HardFilter-PostRollback (see Appendix D.3)**. These two baselines isolate a control interface under the same CBF formulation, allowing us to compare projection-based conrrection against hard filtering methods. We refer the review to Appendix D.3 for the detailed discussion.
> > > To address the concern about baseline strength, we follow reviewer's suggestion to further **tune the logit-steering method** over $\beta \in \{0.5, 1.0, 2.0, 4.0\}$ (see Table 9). We find that RBCBF still performs better after tuning. Specifically, RBCBF achieves $h(x_F)=+0.18$, and $\Delta h_x=5.52$, whereas the best tuned logit-steering variant shows lower terminal safety margin. We also show that stronger steering has limited benefit. Although $\beta=4.0$ yields the largest $\Delta h_x$, its $h(x_F)$ becomes negative again ($-0.06$), showing that aggressive steering can improve a proxy safety score. By contrast, RBCBF improves the terminal state and rolls back earlier.
> > >
> > > **Table 9. tuning of β**
> > > |Method|refusal@trig|h(xF)|Δhx|
> > > |--|-|-|-|
> > > |RBCBF|87.0%| +0.18|5.52|
> > > |LS β=0.5|30.7%|−0.10|3.19|
> > > |LS β=1.0|31.1%|−0.08|3.49|
> > > |LS β=2.0|36.0% |+0.05|3.56|
> > > |LS β=4.0|23.3%|−0.06|3.88|
> > >
> > > **4 (Q4)**
> > > We have reported the **actual wall-clock overhead and the dependence on evaluator frequency of running KL-CBF control in response to Reviewer ZLtM (W2 & Q2: Wall-clock)**. We have also provided results of **how it scales with the decoding length and model size in Table 10.** Our findings are: (i) the overhead ratio is approximately length-invariant, remaining near 1.14× regardless of output length, and (ii) replacing the 7b scorer with a 0.5b scorer reduces overhead as expected, from 1.14× to ~1.01×. We would be happy to clarify further if there is any specific unsolved point.
> > > **Table 10. Wall-Clock Overhead of RBCBF (extended)**
> > >
> > > |Dimension|Condition|sec/prompt|Overhead|Scorer calls/gen|
> > > |-|-|-|-|-|
> > > ||Uncontrolled|1.95|1.00×|0|
> > > ||RBCBF stride=8|2.23|1.14×|15.5|
> > > |Evaluator freq|RBCBF stride=1|6.39|3.28×|124.3|
> > > |Decoding length max=256| RBCBF stride=8|—|≈1.14×|26.5|
> > > |Size|RBCBF scorer=0.5B|~1.97|~1.01×|16.0|
> > >
> > > **5 (Q5)**
> > > We clarify that risk concentration in RBCBF is **more principled than heuristic rollback rules**. The key distinction lies in whether the rollback point is directly specified by a local rule. In heuristic methods, the rollback position is tied to a trigger-adjacent rule, such as the detection time itself, an arbitrary buffer sizes, or another local signal. In RBCBF, the trigger condition only decides whether to intervene, while the rollback position is determined separately by localizing where CBF violation mass has accumulated along the prefix trajectory.
> > >
> > > We further support our claim by **comparing with common heuristics**. **First**, we show that our rollback localization is not a trigger-adjacent or fixed-window heuristic. If it were, rollback positions would follow a concentrated pattern. Instead, they span $0$ to $107$ tokens with $\mathrm{CV} = 1.32$ across all triggerd sequences, indicating a dispersed temporal pattern. **Second**, we show that our method is not a timing-based heuristic but yields meaningful semantic locations. If it were timing-based, the risk state would vary across generators with different tokenization and generation dynamics. Instead, the variation of $h_x(t^*)$ across generators is only $0.30$ (see Table 2 in our response to Reviewer Pdb7), indicating that our rollback position aligns with the similar semantic risk states rather than abitrary timing. Together, these suggest that RBCBF can truly capture content-driven rollback points rather than being heuristic rules.

---

### Official Review · Reviewer_rxfS · 2026-03-10

**Soundness:** 3
**Presentation:** 2
**Significance:** 3
**Originality:** 3
**Overall Recommendation:** 4
**Confidence:** 3

**Summary:**

The paper investigates rollback-based decoding safety The author tries to address two question in decoding safety: (1) when should we rollback, and which prefix step is most influential for the final safety violation; and (2) how to impose corrective constraints to prevent recurrence after rollback?

The author proposes RBCBF to address these two questions, RBCBF uses (1) rollback point identification guided by concentrated violation signals of CBF forward invariance constraints and (2) distribution-level correction enforced through CBF constraints. Empirical results demonstrate RBCBF is effective in reducing harmful outputs while maintaining output quality and correction cost.

**Compliance With Llm Reviewing Policy:**

Affirmed.

**Final Justification:**

I keep my score as weak accept, as the author's rebuttal fully address my comments. I think the paper is a theoretically grounded paper, but it can be further improved by better organizing the experiment presentation and add more ablation study like scorer dependency and time complexity.

**Key Questions For Authors:**

- In lines 156–167, how do you determine the soft weighting function?

- In lines 249–250, what if (Q_t) in Eq. 13 is an empty set? I see later you use slack-augmented projection (Eq. 15) to keep things feasible, but then is Eq. 13 mainly for analysis/intuition?  Did  you check its feasibility? If yes, what’s expected behavior when the hard set is empty?

- Can you describe how these two baselines -- Logit-Steer and Introspect work?

- Do you have different ablation results using other scorers?

**Limitations:**

Yes.

**Strengths And Weaknesses:**

Strengths



- The paper is theoretically grounded and the mathematical modeling makes sense to me (e.g., CBF framing, etc.).

- The idea of turning rollback targeting into a trajectory-level decision is sound and novel.

- The paper is well-written in clearly organized (esp. in sections 2 and 3)



Weaknesses



- Most of the experimental details are deferred to Appendix, making the experiment section hard to understand.

- The reliance on an external scorer is a big dependency. The author uses Qwen2-0.5B-instruct as a risk scorer. I am wondering if the authors have results on other scorers as well.

- I’m a bit concerned about the time complexity of the whole algorithm. In particular, scanning prefixes and running safety scoring for each generation step is high in actual deployment. It can limit the practical usage of the method. Can you elaborate more here?

---

> ### Author Rebuttal · Authors · 2026-03-31
>
> We thank the reviewer for the insightful comments and for recognizing the theoretical grounding and mathematical modeling.
>
> **1. (W2 & Q4: scorer dependence )**
>
>  To address this concern, we add **Table 1**, where we keep the generator fixed and add more scorers. Across these scorers, we find that RBCBF continues to improve terminal safety, although the absolute trigger rate changes with scorer. This reveals that the main effectiveness trend of RBCBF is not tied to one specific scorer. In other words, stronger scorers do affect calibration, but they do not overturn the main conclusion of the framework. For complementary evidence on RBCBF generalization, see our response to **Reviewer Pdb7 (W1 & W4), including Table 2 (generator generalization)** and **Table 7 (scorer noise robustness).**
>
> **Table 1. Scorer Generalizability**
>  *WildJailbreak adversarial set*
> | Safety Scorer        | Trigger Rate | $\Delta h_x(t^* \to F)$ | refusal@triggered |
> |:---------------------|------------:|---------:|------------------:|
> | Qwen2.5-7B    | 91.2%       | +5.18    | 98.6%             |
> | Qwen2-0.5B           | 66.7%       |  +2.24      | 90.2%             |
> | LLaMA-3.1-8B         | 85.0%       | +1.77    | 90.5%             |
> | Mistral-7B-v0.3      | 75.0%       | +4.60    | 86.7%             |
>
> **2. (Q1: Soft weighting function)**
>
> The soft weighting function $w_t$ maps each prefix position to a risk contribution proportional to its CBF deficit below the safety threshold:
>
> $$w_t \=\ \frac{\max\\bigl(0,\ \varepsilon - h(x_{1:t})\bigr)}{\sum_{s=t_0}^{T} \max\\bigl(0,\ \varepsilon - h(x_{1:s})\bigr)}$$
>
> For those positions where the barrier state remains in the safe region (e.g. $h(x_{1:t}) \geq \varepsilon$) receive zero weight. In contrast, positions with high deficits carry proportionally larger weight. The form of $w_t$ follows from the forward invariance property of CBF. The barrier deficit is the natural measure of how badly the trajectory has violated the constraint. We will add an explicit formula for $w_t$ in the revision.
>
> **3. (Q2: Empty set of $Q_t$ in Eq. 13)**
>
> It is true that $Q_t$ (the hard feasible set in Eq. 13) can be empty. Under top-$k$ or top-$p$ truncation, the vocabulary subset may lie entirely in the unsafe region if each token satisfies $h(x_{1:t+1}) < h(x_{1:t}) - \varepsilon$. In this case, Eq. 13 has no solution. We mainly use Eq. 13 to define the ideal hard projection without slack. When $\mathcal{Q}_t \neq \emptyset$, the controller solves Eq. 13 exactly and selects the token in the feasible set to minimize the KL divergence. Eq. 15 serves as a fallback that is activated **only when** $\mathcal{Q}_t = \emptyset$:
>
> $\min_{q \in \Delta(V_t)} D_{KL}(q, p_{base}) + \mu \sum_v q(v)\max(0,\varepsilon - h(x_{1:t}+v))$.
>
> This selects the token that minimizes the barrier violation and yields the least unsafe option. Under the formulation in Eq. 15, the solution is always well-defined (as we proved in Appendix A.4.3), so Eq. 15 guarantees feasibility through a soft penalty. The trajectory may fail to satisfy $h(x_{1:t+1}) \geq h(x_{1:t}) - \varepsilon$ in a single step, but later steps under $\mu$-penalization continue to push it toward the safe region. In practice, this case arises infrequently, typically at the first 1--3 rollback steps when the base model has a strong trend toward a harmful completion. The forcing mechanism further reduces reliance on Eq. 15 by biasing the base distribution before the projection is applied.
>
> **4. (Q3: Baseline explanations)**
>
> The first baseline, Logit-Steer, is a form of soft intervention. Compared with RBCBF, which activates rollback at a risk-concentrated point, Logit-Steer does not perform rollback. Instead, because it does not change the prefix history or apply hard correction to the current trajectory, it only adjusts the logit distribution by adding a vector aligned with the safe direction. We use this direction following the idea in [r1].
> Second, for Rollback-Introspect, we keep the rollback flow the same as RBCBF to make the comparison fair. That is, once the trigger point  $t^* $  is identified, the model first rolls back to that prefix. Then, after rollback, it appends a fixed language template to the safe prefix, and regenerates from $x\_{1:t^*} + \texttt{template}$, without using any CBF projection or constrained token selection. For the detailed template design, we follow the setting in [r2]. We defer a detailed discussion to our response to **Reviewer uWVe (W3 & Q3)**.
>
> **5. (W3: Time complexity)**
>
> To address this, we provide a wall-clock analysis in our response to **Reviewer ZLtM (W2 & Q2: Wall-clock)**. The key result is that RBCBF adds only 1.14× wall-clock overhead in practice.
> References
>
> [r1] Tianlin Liu et al. Decoding-time realignment of language models. arXiv preprint arXiv:2402.02992, 2024.
>
> [r2] Xiaomeng Hu et al. CARE: Decoding time safety alignment via rollback and introspection intervention. arXiv preprint arXiv:2509.06982, 2025.

---

> > ### Author Rebuttal · Reviewer_rxfS · 2026-04-01
> >
> > My concerns are solved. Therefore, I will keep my score.

---

> > > ### Author Response · Authors · 2026-04-04
> > >
> > > We thank the reviewer for confirming that all concerns have been addressed. We appreciate your time and consideration.

---

### Official Review · Reviewer_ZLtM · 2026-03-13

**Soundness:** 3
**Presentation:** 3
**Significance:** 3
**Originality:** 2
**Overall Recommendation:** 4
**Confidence:** 4

**Summary:**

This paper proposes RBCBF (Rollback-based Control Barrier Function), a decoding-time safety intervention framework for Large Language Models. The key innovation is applying Control Barrier Functions (CBFs) to provide a principled formulation for: (1) selecting rollback points based on risk concentration analysis, and (2) performing distribution-level corrective control via KL projection during regeneration.
The authors strive to address a core challenge in decoding-time safety interventions: existing rollback methods rely on heuristic signals rather than principled risk aggregation. Overall, the authors consider a notable topic at the intersection of AI safety and control theory.

**Compliance With Llm Reviewing Policy:**

Affirmed.

**Final Justification:**

Author has addressed my primary concerns and replied the novelty of their work, which I think it's fair enough. The overall recommendation is weak accept.

**Key Questions For Authors:**

1. Variance Estimates: How many random seeds were used? What is the variance in D_term and W across runs? This is essential for assessing statistical significance of improvements.
2. Wall-Clock Latency: What is the actual per-token latency overhead of RBCBF compared to standard decoding? The C_TV proxy measures distribution shift, not time cost.
3. Scorer Error Sensitivity: How does RBCBF performance degrade when the safety scorer has accuracy below 90%? Have you tested with noisier scorers?

**Limitations:**

yes

**Strengths And Weaknesses:**

Strengths:
1. Mathematically Rigorous Framework: The risk aggregation formulation (viewing terminal violations as accumulated negative drift along the prefix) is grounded in control theory. Theorem 1 provides feasibility guarantees for the KL projection, and the α-mass span window is formally justified.
2. Comprehensive Baseline Comparison: The paper compares against 5 methods: LOGIT-STEER, ROLLBACK-INTROSPECT, HardFilter-AlwaysOn, HardFilter-PostRollback, and the concurrent Miyaoka & Inoue 2025 CBF approach. Section D.3 directly compares with Miyaoka's hard-filtering CBF, showing RBCBF achieves lower terminal deficit (0.05 vs 0.12) with smaller distribution shifts (0.96 vs 5.64).
3. Systematic Ablation Studies: The paper includes thorough ablations on correction window size N (20-50), forcing and slack parameters (2×2 ablation in Table 3), and sensitivity to temperature/nucleus sampling.
4. Well-Documented Limitations: The paper includes an Impact Statement discussing social value, and explicitly acknowledges limitations: scorer dependency, bias/coverage gaps, vulnerability to escape attacks, and distribution shift instability.

Weaknesses:
1. No Variance Estimates: Results report single-point metrics (e.g., D_term = 0.539, W = 0.495) without confidence intervals, error bars, or variance across multiple runs. Without knowing result variability, statistical significance of improvements cannot be assessed.
2. No Wall-Clock Latency Analysis: While the paper uses C_TV as a distribution-shift overhead proxy and reports Cost metrics, it doesn't provide actual wall-clock latency measurements. For real-time deployment, per-token latency overhead is critical information.
3. Safety Scorer Error Propagation Not Analyzed: RBCBF relies critically on the safety scoring model h(·), but the paper doesn't analyze how scoring errors propagate to intervention performance or what happens when the scorer is inaccurate.

---

> ### Author Rebuttal · Authors · 2026-03-31
>
> We thank the reviewer for the insightful comments and for the appreciation of the mathematical framework and experiments.
>
> **1. (W1 & Q1 Variance Estimates)**
>
> To rule out seed effects, we ran a multi-seed analysis with seeds 2026, 2027, and 2028, and the results are shown in Table 6. Since seed randomness mainly perturbs individual generation trajectories, it mainly affects trigger timing and intervention strength. Here we report the variance of metrics for trigger and correction. In particular, $\Delta h(t^* \to F)$ measures the safety improvement at which stochastic effects act.
>
> **Table 6. Multi-Seed Variance Analysis**
> *Scorer & generator: Qwen2.5-7B-Instruct, WJB adversarial set.* *The third metric is measured from the trigger point to sequence end.*
> | Metric                     | Seed 2026 | Seed 2027 | Seed 2028 | Mean ±        | CV     |
> |---------------------------|-----------|-----------|-----------|---------------|--------|
> | Trigger rate              | 91%       | 94%       | 94%       | 93.6% ± 2.3%  | 2.5%   |
> | Refusal rate@triggered    | 98%       | 98%       | 97%       | 97.7% ± 0.5%  | 0.5%   |
> | Δh(t\*→F)                 | 5.4       | 6.1       | 5.9       | 5.87 ± 0.46   | 7.8%   |
> | Mean $t\_u$         | 9.8       | 10.9      | 10.5      | 10.52 ± 0.78  | 7.4%   |
>
> All four metrics are stable across seeds. First, the trigger rate is 93.6% ± 2.3% (CV = 2.5%), which is reasonable since the activation frequency does not depend on seeds. Second, we verify the stability of correction quality, with a refusal rate@triggered of 97.7% ± 0.5% (CV = 0.5%). Third, $\Delta h(t^*\rightarrow F)$  is 5.87 ± 0.46 (CV = 7.8%), indicating consistent safety improvement. Fourth, the mean $t\_u$ is 10.52 ± 0.78 tokens (CV = 7.4%), indicating stable detection timing. This pattern aligns with the RBCBF design. Seed randomness mainly affects the generation trajectory and may slightly shift the trigger point.
>
> **2. (W2 & Q2 Wall-clock)**
>
> To address this concern, we provide **Table 4** with actual wall-clock measurements.
>
> **Table 4. Wall-Clock Overhead of RBCBF**
> *Scorer & generator: Qwen2.5-7B-Instruct, WJB adversarial set*. Note: *Scorer calls/gen* denotes the average number of scorer forward passes per generation.
>
> | Condition                | sec/prompt | ±std | tok/s | Scorer calls/gen | Overhead |
> |--------------------------|-----------:|-----:|------:|-----------------:|---------:|
> | Uncontrolled (no scorer) | 1.95       | 0.95 | 63.8  | 0                | 1.00×    |
> | RBCBF stride = 8         | 2.23       | 0.21 | 55.8  | 15.5             | 1.14×    |
> | RBCBF stride = 1         | 6.39       | 0.60 | 19.5  | 124.3            | 3.28×    |
>
> Our main conclusion is that RBCBF adds only **1.14× wall-clock overhead** in practice. The average latency increases from **1.95 sec/prompt** to **2.23 sec/prompt**, which is an absolute increase of **0.28 s**. Compared with uncontrolled generation, it also shows lower variance (**std = 0.21**). This result mainly comes from our design choices at inference time. The explanation is twofold.
>
> First, we score the prefix every **8 tokens**, rather than once per token. As a result, for a 128-token generation, the number of scorer calls is reduced from about **128** to about **16**. We consider this design reasonable because the safety margin usually does not change sharply after only a few new tokens. In practice, the scorer becomes more informative after a sentence fragment has formed. We also isolate the cost of per-token scoring through a **stride=1 ablation**, where the wall-clock overhead increases to **3.28×**.
>
> Second, we run the experiments on **two A100 GPUs**. The generator and the scorer are deployed on separate GPUs and execute in parallel. Under this setting, an **8× reduction** in scorer calls corresponds to a **2.9× reduction** in wall-clock overhead. For a single-GPU setup, we expect the overhead to lie between **1.14×** and **3.28×**.
>
> For reference, prior decoding-time safety methods based on per-token scoring, such as **SafeDecoding** [r1] and **DExperts** [r2], require about **128 scorer calls per generation** and report about **2× overhead**, while **RAIN** [r3] performs serial self-evaluation rewrites and reports about **10× overhead**. In comparison, RBCBF uses only **15.5 scorer calls per generation**.
>
> References
>
> [r1] Zhangchen Xu et al. SafeDecoding: Defending against Jailbreak Attacks via Safety-Aware Decoding. ACL, 2024.
>
> [r2] Alisa Liu et al. DExperts: Decoding-Time Controlled Text Generation with Experts and Anti-Experts. ACL-IJCNLP, 2021.
>
> [r3] Yuhui Li et al. RAIN: Your Language Models Can Align Themselves without Finetuning. ICLR, 2024.
>
> **3. (W3 & Q3 Scorer Error Sensitivity)**
>
> To address this concern, we present a scorer-noise robustness study in **Table 7**. Please see our detailed response under **Reviewer Pdb7**, where we report the results with noisier scorers and discuss how scorer errors propagate to RBCBF performance.

---

> > ### Author Rebuttal · Reviewer_ZLtM · 2026-04-05
> >
> > Thank you for the detailed rebuttal; the multi-seed variance analysis, wall-clock latency measurements, and scorer noise robustness results have addressed my primary concerns. Meanwhile, I still find the originality somewhat limited given the existing rollback paradigm and CBF literature in safe RL, and I am maintaining my score.

---

> > > ### Author Response · Authors · 2026-04-06
> > >
> > > Thank you for confirming that the primary concerns are addressed, and we sincerely appreciate the reviewer’s careful reading and insightful comments.
> > >
> > > We agree that there has been related work on both rollback and safety control based on CBF in safe RL. We would like to kindly explain the core novelty of RBCBF from **three parts**.
> > >
> > > **First**, RBCBF itself is not an incremental combination of the two components. Its novelty mainly lies in the principled formulation. Due to the forward-invariance property of discrete-time CBF, terminal violation can be formed as accumulated risk along the prefix, and this trajectory-level risk concentration is then used to infer rollback points.
> > >
> > > **Second**, compared with existing rollback methods, which typically choose rollback points from local signals, RBCBF addresses which earlier prefix step contributed most to the terminal violation based on a **novel risk concentration view**. In this sense, RBCBF provides a more principled way to localize where the unsafe state is concentrated based on CBF theory.
> > >
> > > **Third**, compared with CBF-based safe control in RL, which typically focuses on stepwise control, RBCBF addresses a specific challenge in LLM decoding. During the generation process, once a prefix has been realized, the feasible space of its suffix becomes constrained. As a result, safety control cannot just focus on constraining the next decoding step. At the same time, it also needs to determine whether the unsafe state is concentrated in the realized prefix. Here, we use RBCBF to address these two coupled questions simultaneously. **The core insight** is that the **same terminal violation anchor from the forward-invariance property is carried into post-rollback correction based on KL-CBF**, so rollback localization and suffix control become two coupled components in one framework.
> > >
> > > We therefore believe the contribution of RBCBF lies in a more principled and unified formulation for rollback localization and post-rollback correction while also providing a new view for understanding how unsafe outputs are formed during LLM decoding, rather than a simple stacking of rollback and CBF, especially for **safe LLM decoding**. We hope this explanation could better clarify the **originality of RBCBF** that we did not explain clearly enough before.

---

### Official Review · Reviewer_Pdb7 · 2026-03-15

**Soundness:** 3
**Presentation:** 3
**Significance:** 4
**Originality:** 4
**Overall Recommendation:** 4
**Confidence:** 4

**Summary:**

This paper presents RBCBF, a novel rollback-based decoding-time safety alignment framework designed to address the limitations of existing decoding-time safety interventions. The framework leverages risk aggregation and Control Barrier Function (CBF) theory to select intervention steps and apply distribution-level corrective control, thereby reducing harmful responses and lowering the recurrence rate of safety violations.

**Compliance With Llm Reviewing Policy:**

Affirmed.

**Final Justification:**

Weak accept.

**Key Questions For Authors:**

See weakness.
Despite the above concerns, I appreciate the novelty of this work and find several of its design choices genuinely elegant — particularly the risk concentration perspective and the KL projection formulation under multi-rule CBF constraints. I would be willing to raise my score if the authors can provide clarifications and additional evidence addressing my concerns about the experimental setup.

**Limitations:**

RBCBF offers a theoretically grounded framework for decoding-time safety alignment, and the risk concentration perspective is a valuable and original contribution. However, there remains a notable gap between the theoretical rigor of the CBF-based formulation and its demonstrated practical effectiveness. The heavy reliance on an external scorer, the inherently weak nature of expectation-only safety guarantees, and the limited scope of experimental validation collectively suggest that this work is, at present, closer to a proof of concept than a deployment-ready solution. Closing this gap will require more comprehensive evaluation and stronger empirical grounding.

**Strengths And Weaknesses:**

### Strength:
1. RBCBF provides a principled theoretical foundation for rollback and intervention by combining risk aggregation with CBF theory. This offers a fresh perspective on understanding and improving the safety of language models, and has the potential to contribute to the development of more reliable AI systems.

2. The framework is designed to be flexible and adaptable to diverse safety requirements, suggesting broad applicability across different deployment scenarios.

### Weakness:
1. The safety scorer used in the experiments is relatively small. It remains unclear whether a larger and more capable scoring model would lead to meaningfully different results. The authors should discuss or empirically evaluate how scorer quality affects the overall framework performance.

2. Insufficient ablation on key hyperparameters: The ablation studies do not adequately cover several critical hyperparameters. Notably, the slack penalty μ which directly governs the feasibility-correction trade-off in the KL projection — is never assigned a clearly reported value, and no sensitivity analysis is provided for it. This makes it difficult to assess the robustness of the proposed method.

3. Cost metric does not reflect real-world latency. The unified cost metric C defined in the paper is a normalized composite index that aggregates various components. However, it does not correspond to actual inference latency or FLOPs, and therefore cannot directly inform practitioners about the real-world overhead or user-facing response delay introduced by the framework.

4. Limited experiments: The evaluation is conducted on a single dataset (WildJailbreak) with a single base model (Qwen2.5-7B-Instruct). This narrow experimental setup makes it difficult to assess the generalizability of RBCBF across different data distributions, attack types, and model architectures. A broader evaluation covering additional datasets, stronger baselines, and diverse model families would significantly strengthen the empirical claims.

---

> ### Author Rebuttal · Authors · 2026-03-31
>
> We thank the reviewer for the insightful comments and for recognizing the theoretical perspective and practical applicability of the approach.
>
> **1. (W1 & W4)**
>
>  We provide evidence from **three parts**: **(i)** cross-scorer generalizability, **(ii)** cross-generator generalizability, and **(iii)** scorer-noise robustness.
>
> * First, for **cross-scorer generalizability**, we refer the reviewer to our response under **Reviewer rxfS: 1. (W2 & Q4: scorer dependence)**.
>
> * Second, we test **cross-generator generalizability**, i.e., whether RBCBF depends on one specific generator, as shown in **Table 2**. We keep the safety scorer fixed and replace the generator with different models. Across these generators, we find that RBCBF is capable of generalizing and continues to improve terminal safety. Concretely, all three generators show an upward trend in safety value after rollback, with $\Delta h_x(t^* \to F)$ staying in the range of +1.46 to +2.69. The terminal margin typically moves from an unsafe state at $t^*$ back toward the safe region at the end.
>
> **Table 2: Generator Generalizability**
> *Scorer: Qwen2-0.5B-Instruct, WildJailbreak adversarial set.*
>
> | Generator | $h(x_0)$ | $h(x_{t^*})$ | $h(x_F)$ | $\Delta h_x(t^* \to F)$ | Safety@triggered |
> |-------------------|-------|-----------|-------|----------|------------------|
> | Qwen2.5-7B        | +2.44 | −1.24     | +0.70 | +1.94    | 83%              |
> | LLaMA-3.1-8B      | +1.03 | −1.47     | −0.02 | +1.46    | 87%              |
> | Mistral-7B-v0.3   | +2.17 | −1.54     | +1.15 | +2.69    | 80%              |
>
> * Third, we test **scorer-noise robustness** with a standard perturbation model, as shown in **Table 7**. Specifically, we inject independent Gaussian noise into the scorer logits, i.e., $\mathcal{N}(0,\sigma^2)$, where $\sigma \in \lbrace 0.25,0.50,1.00\rbrace$. This follows standard uncertainty at the logit level. Our main finding is that RBCBF remains stable under scorer noise. Across the tested noise levels, the **trigger rate** stays within **95.1%--97.1%** and $\Delta h_x(t^* \to F)$ remains within **4.93--5.30**, with no sign of monotonic collapse. This indicates degradation rather than failure. We attribute this stability to structural factors. First, as discussed in Appendix C, RBCBF activates only when two continuous observations fall below the threshold. For unsafe trajectories, the observed margin at the trigger point is already far below the threshold (e.g., $h(t^*) \approx -9.5$ while $\epsilon=-0.5$), so we verify the robustness of RBCBF under standard noise. We hope these results could address the reviewers' concern.
>
> **Table 7. Scorer Noise Robustness of RBCBF**
>
> *Scorer & generator: Qwen2.5-7B-Instruct, WJB adversarial set*
>
> | Metric | $\sigma=0.00$ | $\sigma=0.25$ | $\sigma=0.50$ | $\sigma=1.00$ | Change vs. $\sigma=0$ |
> |:--|--:|--:|--:|--:|--:|
> | Trigger rate | 0.936 ± 0.023 | 0.971 | 0.951 | 0.970 | +3.4 pp |
> | Refusal@triggered | 0.977 ± 0.005 | 0.974 | 0.959 | 0.958 | −1.9 pp |
> | $\Delta h(t^*\rightarrow F)$ | 5.87 ± 0.46 | 5.09 | 4.93 | 5.30 | −0.57 (−9.7%) |
> | Mean $h(x_{t^*})$ | — | −8.96 | −9.54 | −9.60 | — |
> | Mean $h(x_F)$ | −3.29 | −3.88 | −4.61 | −4.30 | — |
>
> **2. (W2)**
>
> To address this concern, we further report a sensitivity analysis across $\mu \in \lbrace 5, 10, 15, 20 \rbrace$, as shown in Table 8. We first report that the trigger rate stays stable across all $\mu$ values. This is because $\mu$ is decoupled from the trigger condition and mainly plays the role of a penalty coefficient in Eq. 15. We also find that safety@triggered increases as $\mu$ increases, from 78.2% at $\mu=5$ to 85.9% at $\mu=20$. Last, both the safety rate and $\Delta h(t^* \to F)$ achieve saturation from $\mu=15$ to $\mu=20$. This is expected because there exists a threshold, beyond which increasing the penalty further only increases the projection cost rather than improving the safety margin. Therefore, we keep this parameter as $\mu=15$ across all experiments.
>
> **Table 8. μ sensitivity**
> *Generator: Qwen2.5-7B-Instruct, WildJailbreak adversarial set.*
> | $\mu$ | trigger_rate | safety@triggered | $\Delta h(t^* \to F)$ |
> |------|--------------|-------------------|------------------------|
> | 5    | 93.5%       | 78.2%              | +4.6                   |
> | 10   | 93.4%       | 83.0%              | +5.4                   |
> | 15   | 93.6%       | 85.2%              | +5.9                   |
> | 20   | 93.5%       | 85.9%              | +6.0                   |
>
> **3. (W3)**
>
> We refer the reviewer to the detailed discussion under **2. (W2 & Q2: Wall-clock)** in our response to **Reviewer ZLtM**, where we further clarify the wall-clock and response delay in real-world deployment.
>
> **4. (W4)**
>
> For a complete response, we refer the reviewer to **1. (Q1 & W2 & W5)** in our response to **Reviewer uWVe** where we provide a broader evaluation over the full distribution.

---

> > ### Author Rebuttal · Reviewer_Pdb7 · 2026-04-04
> >
> > Thanks for the detailed reply, my concerns have been adequately addressed, I will raise my score accordingly.

---

> > > ### Author Response · Authors · 2026-04-04
> > >
> > > Thank you for your time and consideration. We sincerely appreciate the reviewer for raising the score.

---

### Decision · Program_Chairs · 2026-04-30

**Decision:**

Accept (regular)

**Comment:**

This paper develops a decoding-time safety alignment method based on Control Barrier Functions (CBFs), including a new approach for selecting rollback points and safety-constrained sampling. Reviewers appreciate that the proposed method is a principled framework and can potentially be broadly applied across different safety tasks.

During the rebuttal, the authors provided more detailed explanations and new experimental results that addressed most concerns.

Remaining concerns are mainly on novelty, dependency on the scorer, and time complexity.